# Kinome Render: a stand-alone and web-accessible tool to annotate the human protein kinome tree

Matthieu Chartier[1,3], Thierry Chénard[1,3], Jonathan Barker[2] and Rafael Najmanovich[1]

[1] Department of Biochemistry, Faculty of Medicine and Health Sciences, Université de Sherbrooke, Québec, Canada
[2] European Bioinformatics Institute, Wellcome Trust Genome Campus, Hinxton, Cambridge, UK
[3] These authors contributed equally to this work.

## ABSTRACT

Human protein kinases play fundamental roles mediating the majority of signal transduction pathways in eukaryotic cells as well as a multitude of other processes involved in metabolism, cell-cycle regulation, cellular shape, motility, differentiation and apoptosis. The human protein kinome contains 518 members. Most studies that focus on the human kinome require, at some point, the visualization of large amounts of data. The visualization of such data within the framework of a phylogenetic tree may help identify key relationships between different protein kinases in view of their evolutionary distance and the information used to annotate the kinome tree. For example, studies that focus on the promiscuity of kinase inhibitors can benefit from the annotations to depict binding affinities across kinase groups. Images involving the mapping of information into the kinome tree are common. However, producing such figures manually can be a long arduous process prone to errors. To circumvent this issue, we have developed a web-based tool called Kinome Render (KR) that produces customized annotations on the human kinome tree. KR allows the creation and automatic overlay of customizable text or shape-based annotations of different sizes and colors on the human kinome tree. The web interface can be accessed at: http://bcb.med.usherbrooke.ca/kinomerender. A stand-alone version is also available and can be run locally.

# INTRODUCTION

The human genome codes for 518 protein kinases, also known as the human kinome, which represent a little less than 2% of all human genes (*Manning et al., 2002*). These kinases regulate multiple biological processes such as apoptosis, transcription, mobility of the cell and metabolism by catalyzing the covalent bonding of a phosphate group to an amino acid with a free hydroxyl group. The proliferation of cancer cells often involves alterations in kinase activity in signaling pathways. As such, human protein kinases represent important targets in drug design.

Corresponding author
Rafael Najmanovich,
rafael.najmanovich@usherbrooke.ca

Panels of binding assays and bioinformatic analyses related to the human kinome produce large quantities of data, which are hard to analyze without appropriate visualization tools. One way of visualizing data is by annotating the kinome phylogenetic tree that originally accompanied the first analysis of the human kinome (*Manning et al., 2002*). This phylogenetic tree depicts the relationship between different members of the superfamily based on the sequence similarity of their catalytic domains. It became an iconic representation of the kinome due to its high graphic and artistic qualities. Annotated versions of this phylogenetic tree have been used multiple times in the past in articles related to the human kinome (*Karaman et al., 2008*; *Marsden & Knapp, 2008*; *Edwards, 2009*; *Fedorov et al., 2010*). Some figures in these publications were created by hand in a laborious effort (A Edwards, B Marsden, & S Knapp, pers. comm., 2008). For the reason aforementioned, we have created Kinome Render, a tool that can be used via a web interface or that can be downloaded to run locally. KR enables the annotation of the human kinome tree while reducing the amount of work involved and, more importantly, the risk of errors.

At least two other methods exist to annotate the Human Kinome. TREEspot is a commercial online-only service developed by Discoverx (http://www.discoverx.com/) that is geared specifically for the visualization of inhibition binding profiles. TREEspot allows only the plotting of circles whose radii scale with binding affinity overlaying these on a circular dendrogram of the human kinase family. The human kinome Java component (http://tripod.nih.gov/?p=260) is JavaScript that allows annotation of kinases on the human kinome phylogenetic tree. While simple to use, no interface is available. In both cases, the variety of annotation formats is limited. Kinome Render has a very large variety of annotation formats with freely available web interface and stand-alone versions and a large variety of methods to input data.

Kinome Render works by overlaying annotations of specific kinases on a template of the human kinase phylogenetic tree. Annotations, which can be circles, polygons or text of different sizes and colors, are centered at the coordinates corresponding to the leaf of the tree that defines the kinase to be annotated. The annotations can be combined to represent practically any kind of data about particular kinases within the framework of the entire family as represented in the kinome tree.

## THE KINOME RENDER TOOL

### Presentation

In Kinome Render, any number and combination of annotations using text strings and geometric shapes with independent color and scale attributes are overlaid at the position of specified kinases on the human kinome tree. To render the annotated tree, KR ultimately requires a text file written in KR syntax that contains all the information needed to draw the annotations. There are three ways to create an annotated tree using KR (summarized in Fig. 1): (1) by running the stand-alone version locally, (2) by using the web interface directly, or (3) by uploading a KR file through the web interface. Using the stand-alone version requires a text file that lists the kinase(s) with their corresponding annotations as

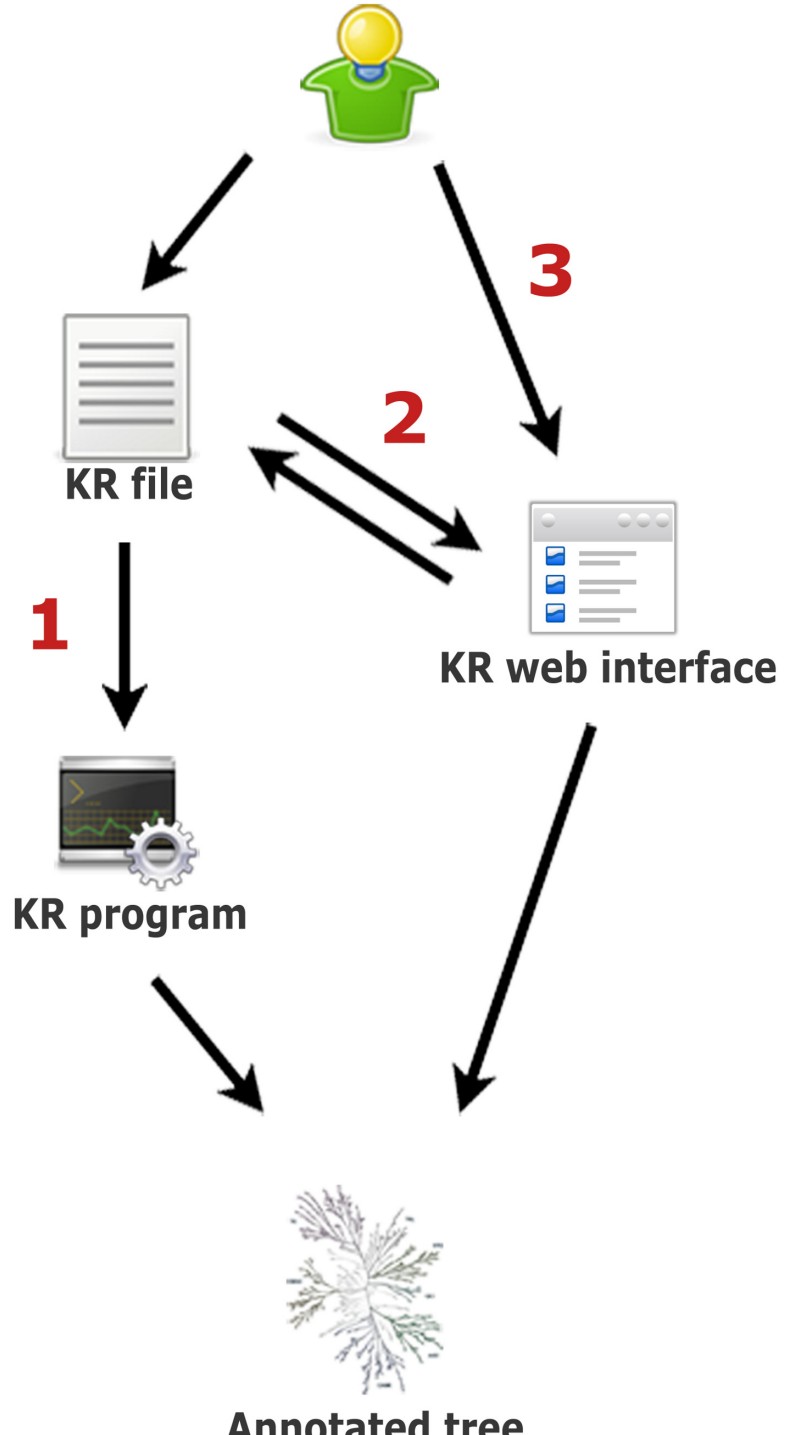

**Annotated tree**

**Figure 1** **Three ways to create an annotated tree with Kinome Render.** (1) Write a KR file and use it as input to the stand-alone version of the program on a local machine. (2) Write a KR file and upload it to the web interface. This allows the editing of the annotations in the file using the web interface as well 

**Peer**J

input. This file must be written in KR syntax, a flexible simple language described below.
As mentioned, the web interface can be used in two ways. First, by creating an annotated
tree from scratch, to help those not familiar either with running stand-alone programs
or the KR syntax. Second, by uploading a file written in the KR syntax, which loads the
annotations encoded in the file to the interface. From there, additional annotations can be
created and existing ones edited. KR files uploaded to the interface must be plain text.

In addition to the annotation of the kinome tree, kinome render permits the creation
of a legend that describes the meaning of the different annotations used. Considering
the vast flexibility of possible annotations, any procedure to automate the creation of
legends would be unnecessarily restrictive. In order to permit the creation of legends
without restrictions, a minimal knowledge of the KR syntax is necessary in both web and
stand-alone versions. However, the creation of a legend is optional and the user can always
create a KR image without a legend and later annotate the image independently with any
image processing software.

Kinome Render utilizes a database containing a total of 523 entries (Table S1) retrieved
from *Manning et al. (2002)*. Each of the 523 entries corresponds to a unique leaf of the
original human kinome tree. From the initial 518 human kinases, 8 atypical kinases (BCR,
FASTK, G11, H11, TAF1, TAF1L, A6, A6r) are absent from the original tree. Additionally,
there are 13 kinases that have two distinct catalytic domains (JAK1, JAK2, JAK3, MSK1,
MSK2, RSK1, RSK2, RSK3, Tyk2, GCN2, RSK4, SPEG, Obscn) and thus are present twice
on the tree (their names are tagged with ~b). A list of corresponding names and synonyms,
Uniprot ID, IPI, group, family, subfamily, full protein sequence and kinase domain
sequence are stored in the database for each entry. The kinase domain sequences of the
atypical kinases are absent in *Manning et al. (2002)* and thus not present in the database.
The reason for compiling the protein sequences is explained below. The procedures to use
KR either via the web interface or as a stand-alone software are described in the following
sections.

## The Kinome Render web interface

The KR web interface contains all the necessary tools to create an annotated human
kinome tree. In the interface, users can add, modify or delete annotations and render
the annotated tree in 5 different formats (PostScript, PNG, JPG, PDF, TIFF). At any time,
the user can save the KR input file containing all the created annotations. This file can
be uploaded in the interface to restore a previous session and make modifications on the
annotations.

In the interface, the kinases are sorted in alphabetical order and the user can browse
through the list to select one or more kinases to annotate. The selection can also be
performed in another three distinct ways. The first method is by searching the database

using any combination of Manning identifiers (name, accession ID, group, family or subfamily) as well as Uniprot ID or International Protein Index (IPI). When searching by name, the query string is matched against the Manning name, the protein's full name (e.g., Mitotic checkpoint serine/threonine-protein kinase BUB1 beta) and a list of synonyms retrieved from the Uniprot database (*UniProt Consortium, 2013*). The second method is by sequence alignment. To do so, the user must input a protein sequence in Fasta format that will be aligned to the full protein sequences of the 523 entries using Fasta (*Pearson & Lipman, 1988*). The results are sorted by E-value and the percentages of similarity and identity are displayed. At any time, the user can hover over the kinase name to display a tooltip containing the kinase's full name and a list of synonyms. It is worth noting that even if the kinome tree is based on kinase domain similarities, users can use sequence attributes located outside the kinase domain to generate annotation. Sequence alignments against the full sequences may be useful for that. The third method, called Batch Upload, allows the selection of kinases based on a list provided by the user. The elements of the list can be either the kinase name (column Manning Name in Table S1), Uniprot ID or IPI. The elements of the list can separated by commas, semicolons or new lines.

Once the desired kinases have been selected, the user can customize the annotation that will be overlaid at the position of the selected kinase(s) on the phylogenetic tree. There are 3 types of annotations: (1) a custom text string (max. 25 characters), (2) a shape, or (3) the kinase's name. The custom text strings or the kinase names can be inside a box or underlined. The shape can be a circle or a polygon of any number of sides, all with 3 different styles: shape outline, color filled shape or color filled shape with a black outline. For all annotation types, the size and color can be adjusted. The color can be selected from a predefined list of colors or by entering RGB values (between 0 and 1). The combination of these annotation styling properties provide interesting data visualization possibilities that can help reveal trends across the entire human protein kinase family (see examples below). The formatting options can be modified or deleted at any time by clicking the modify and delete buttons of any annotation. A legend can also be created on the final image. To do so, the elements of the legend must be written in KR syntax (described in the next section) and inserted in the legend section of the interface.

There is no limit to the number of annotations that can be created, allowing the use of Kinome Render as a tool for multi-dimensional kinase data visualization as the vast number of shapes, colors, scale and text options permit the overlay of an immense set of properties. For example, a user can partition a dendrogram built from any distance or similarity matrix and annotate the members of each cluster in a particular way. While the implementation of such a procedure is beyond the scope of Kinome Render, only minimal knowledge of statistical programs is necessary to generate this type of data. Therefore, Kinome Render should be seen as a means to visualize multidimensional data. Once all desired annotations are created, the user can choose to display the names of all remaining kinases. The names will appear in color black and size 10. This format cannot be changed. If this option is not selected, only the backbone of the tree and the defined annotations

will be displayed. The user has the option to provide a filename and to choose an output format (PDF, JPEG, PNG or TIFF). Note that a PostScript image is generated by default in addition to the chosen format. There are two templates to choose from: one with only the typical kinases tree and a second one with the typical kinases tree accompanied by a small sub-tree with the atypical kinases in the bottom left. If any atypical kinases are used in the annotations, the interface will constrain the selection to the second template.

Once the disclaimer is accepted and the form submitted, a new webpage provides links to download (1) the final image in the user-selected format, (2) the PostScript image, and (3) a KR file. As mentioned above, the KR file can be saved for use with the stand-alone version or uploaded in the interface in order to restore all previously created annotations to edit, add or delete annotations from the tree.

We understand that users might have concerns about the privacy of their data. Some files are stored temporarily in our servers for the proper functioning of the interface. These files are deleted automatically after a 72-h period. We also provide the option to immediately delete all related files.

## Running Kinome Render locally

The KR program behind the web interface is a Perl script that can be used as a stand-alone version. It generates a PostScript file that defines the final image. If other formats are required, this PostScript file should be converted using the appropriate programs. For example, this conversion is done automatically with some programs like Preview (Mac) or Adobe's Acrobat Reader (Windows or Mac). As previously mentioned, the KR file can also be uploaded directly to the web interface in order to render the image in multiple formats. The stand-alone version of the program takes as argument (1) a text file containing the annotations written in KR syntax (described below), (2) the template to be used (if it's not template 1), and (3) the output location and file name. The templates are in the downloadable Kinome Render folder, available for download at http://bcb.med. usherbrooke.ca/kinomerender.

The KR format works with short one-line commands (listed in Table 2). The first obligatory command for all annotations is the "at kinaseName" replacing kinaseName with the proper name (column "Name" in Table S1). This indicates to which kinase assign the annotation. By default, the color is black and the scale is 10. These can be changed using the "color" and "scale" commands. These properties remain unchanged until redefined. The KR program automatically overlays annotations in decreasing scale order so as not to hide smaller annotations in the vicinity of larger ones. It is not necessary to manually order annotations by scale when entering annotations iteratively in the web interface or when using input files in either version.

The annotation type is defined immediately after the position, scale and color have been defined as described above. If the annotation to be drawn is a text, simply write "text string" replacing string with the desired label. Special characters such as Greek letters can be used. For example, to insert an $\alpha$ simply write "[alpha]" (see Table 1 for a list of symbols and their corresponding code). If the text needs to be underlined or boxed, the

**Table 1  Codes for special symbols for use in text string annotations.**

| Code | Symbol | Code | Symbol |
|------|--------|------|--------|
| [alpha] | $\alpha$ | [sigma1] | $\varsigma$ |
| [beta] | $\beta$ | [upsilon] | $\upsilon$ |
| [chi] | $\chi$ | [tau] | $\tau$ |
| [delta] | $\delta$ | [xi] | $\xi$ |
| [Delta] | $\Delta$ | [psi] | $\psi$ |
| [epsilon] | $\varepsilon$ | [Psi] | $\Psi$ |
| [phi] | $\varphi$ | [zeta] | $\zeta$ |
| [gamma] | $\gamma$ | [intersection] | $\cap$ |
| [Gamma] | $\Gamma$ | [union] | $\cup$ |
| [eta] | $\eta$ | [angle] | $\angle$ |
| [iota] | $\iota$ | [equivalence] | $\equiv$ |
| [lambda] | $\lambda$ | [plusminus] | $\pm$ |
| [kappa] | $\kappa$ | [lesserequal] | $\leq$ |
| [mu] | $\mu$ | [greaterequal] | $\geq$ |
| [nu] | $\nu$ | [diamond] | $\blacklozenge$ |
| [pi] | $\pi$ | [heart] | $\heartsuit$ |
| [Pi] | $\Pi$ | [spade] | $\spadesuit$ |
| [rho] | $\rho$ | [club] | $\clubsuit$ |
| [theta] | $\theta$ | | |
| [Theta] | $\Theta$ | | |
| [theta1] | $\vartheta$ | | |
| [omega] | $\omega$ | | |
| [Omega] | $\Omega$ | | |
| [sigma] | $\sigma$ | | |
| [Sigma] | $\Sigma$ | | |

commands "underlined" or "boxed" must be written immediately on the following line. Shapes (circle or polygon) can also be drawn with three different styles: (1) a shape's outline, (2) a color-filled shape or (3) a color-filled shape with black border. The syntax for each shape type is listed in Table 2. Some examples on how to use these commands appear below.

KR also allows the creation of legends. To do so, the user has to write the one-line command "legend" and, following this line, add the information of each element of the legend in the same format as the annotation. The "next-line" command can be used to jump to the following line of the legend. The "space" command can be added to concatenate a space at the current position. The legend can be enclosed in a box by adding the "legendBox" command at the very end of the legend. Refer to section below for an example on how to use legends.

Finally, if the user wants all remaining kinases to be labeled with their respective name in black, size 10, the line "remainder" must be added at the end of the text file.

**Table 2** Description of Kinome Render formatting syntax language commands.

| Command | Description | Value |
|---|---|---|
| at (value) | Specifies the kinase to which to assign the annotation | Name (see Table S1) i.e., Abl |
| color (value) | Sets the color of the annotations | RGB values between 0 and 1 i.e., 1 0.67 0.5 or one of the predefined color (black, red, green, blue, yellow, magenta, cyan, orange, pink, grey) |
| scale (value) | Sets the size of the annotations | Integer (default 10) |
| text (value) | Prints a string of text | String (max. 25 characters, see Table 1 for special characters) |
| circle<br>circle-filled<br>circle-lined | Draws a circle<br>Draws a color-filled circle<br>Draws a color-filled circle with a black outline | N.A. |
| polygon (value)<br>polygon-filled (value)<br>polygon-lined (value) | Draws a polygon of (value) sides<br>Draws a color-filled polygon<br>Draws a color-filled polygon with a black outline | integer (e.g.,: polygon 4 will draw a square) |
| boxed | Encloses the annotation in a box | N.A. |
| underlined | Underlines the annotation | N.A. |
| remainder | Prints the names of all non-annotated kinases (add this command once at the last line of your file) | N.A. |
| legend | Declares that the following lines describe the legend | N.A. |
| space | Adds a trailing space in an element of the legend | N.A. |
| next-line | Jumps to the next line in the legend | N.A |
| legendBox | Inserts a box around the legend; Must be placed in the last line of the legend section | |

## EXAMPLES

Below are some examples that show how to create basic and advanced figures with Kinome Render. The plain text KR input files for each of the examples are available for direct download through the Kinome Render web interface. These input files can be used as input for the stand-alone and for the web interface.

### Ex.1 - Quick start

This example shows the basics of the KR syntax (Example File 1). We will annotate the PINK1, MPSK1 and BIKE kinases, each with a different color, size and annotation type.

*at PINK1*
*color 1 0 1*
*scale 30*
*polygon-filled 3*

```
at MPSK1
color 0.92 0.08 0.08
scale 20
circle-lined

at BIKE
color 0 0.67 0
scale 10
text BIKE
```

A plain text file submitted with the above code will yield what we see in Fig. 2. Note that the whole kinome tree is rendered in the final figure but only a fraction is shown in Fig. 2 for simplicity.

### Ex.2 - Draw legends

For this example, we will draw a legend for the example above. The code below yields a legend (Fig. 3). Again, the whole tree is generated but only the legend is shown in Fig. 3 for simplicity. Example File 2 contains the code of example 1 concatenated to example 2.

```
legend
scale 20

color 1 0 1
polygon-filled 3
space
text PINK1 is annotated with a pink triangle

next-line
color 0.92 0.08 0.08
circle-lined
space
text MPSK1 is annotated with a red circle

next-line
color 0 0.67 0
text BIKE is annotated in green

legendBox
```

One interesting thing to note in the last example is that the scale was set only once at the beginning. This will make all objects declared afterwards render at scale 20. The same thing is applicable to the color. This is a general property of the KR language and is applicable to both the kinome annotations and legend. For example, if we want all annotations or legend elements of our tree to be in orange, we can declare once at the beginning of our code "*color 1 0.65 0*". Predefined colors (see color section in Table 2) can also be used. For example "*color orange*" will work as well. Note that kinases annotated with the "*remainder*"

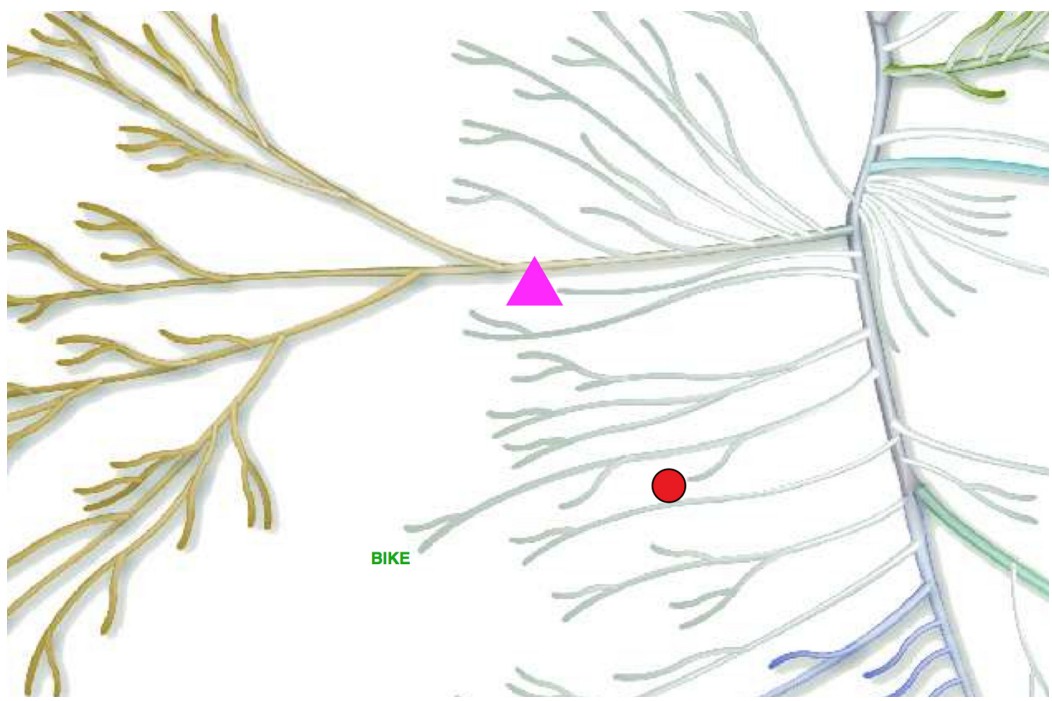

**Figure 2 Example of a simple annotated tree created by Kinome Render.** PINK1, MPSK1 and BIKE kinases labeled using different annotation types.

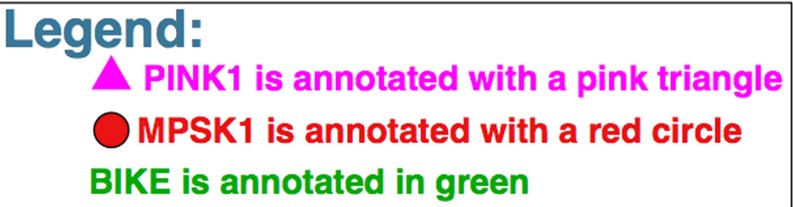

**Figure 3 Example of a legend created by Kinome Render.** A legend example for Fig. 2 to illustrate how to create legends.

command are not affected by these changes; namely, remaining kinases will be displayed using their names in black size 10. The "*next-line*" command is used between each element of the legend. The "*space*" command is used to concatenate a space between two elements on the same line of the legend. The command "*legendBox*" is used at the very end of the legend to draw a box around the legend.

**Advanced example 1**

In this example, we combine information about the availability of PDB structures of kinase domains with the list of kinases studied by *Karaman et al. (2008)* (Fig. 4). We use a color code to distinguish three possibilities: (1) protein kinases studied by Karaman et al. (blue), (2) protein kinases with a PDB structure representing the kinase domain (red), and (3)

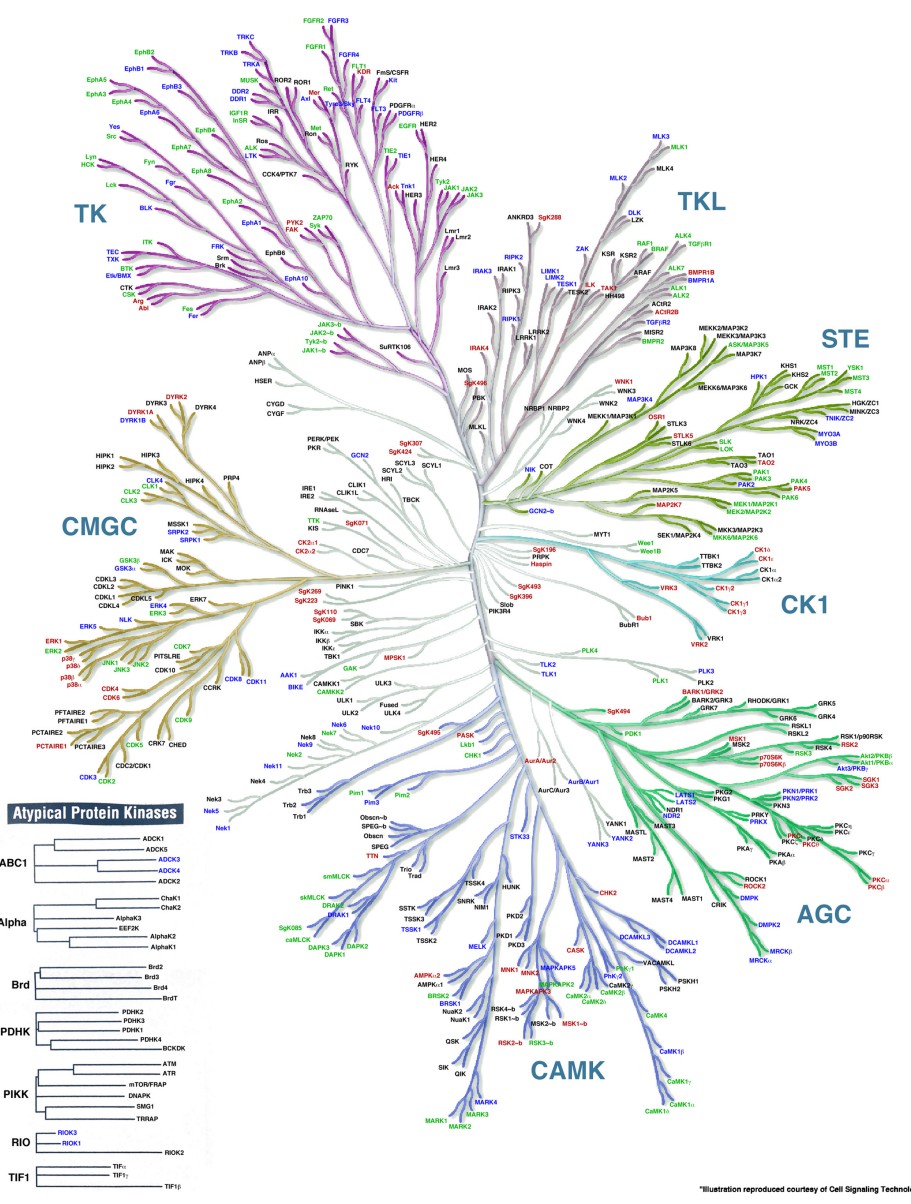

**Figure 4 Kinome Render can be applied to any kind of data, not just binding affinities.** A phylogenetic tree created by Kinome Render showing protein kinases studied by *Karaman et al. (2008)* (blue), protein kinases with a PDB structure representing the kinase domain (red) and protein kinases both studied by Karaman et al. and with a PDB structure (green).

protein kinases both studied by Karaman et al. and with a PDB structure (green). The rest of the kinases are annotated in black using the "remainder" command. The KR input file used to render this figure is Example File 3.

## Advanced example 2

The study of *Karaman et al. (2008)* contains many figures that use the human kinome tree to depict the promiscuity (cross-activity) of a variety of drugs. In these figures, the size of the circles represent the affinity of the drug for a particular protein. Processing Example File 4 with KR yields what we see in Fig. 5. Overlaying binding information on the kinome tree can reveal if the binding profile of a drug is widespread across different families or restricted to close relatives of the intended target. In the specific example used, we can quickly see that the inhibitor Sunitinib is very promiscuous and binds very strongly to members of the TK family.

## Inhibitor binding profiles

The example in the last paragraph shows what may be the most common application for a tool like Kinome Render, namely the visualization of inhibition profiles across the entire human kinase family. We have implemented a special procedure to facilitate the creation of such images for the specific case of inhibitor binding profiles. The procedure, described below, can be used via the web interface or locally by running the script available in the Kinome Render Downloadable folder. Users need to use as input a tab-separated multi-line file. In this file, each line represents a particular kinase and each column an inhibitor. Accordingly, the first row should contain inhibitor names and the first column either kinase names (column "Name" in Table S1), UniprotIDs or IPIs. Any particular cell should contain the $K_d$, $IC_{50}$ or any other numerical measure where lower values mean stronger effects. The user needs to specify the units, either nM or μM, for the legend to be correctly drawn. Furthermore, the user can choose to annotate kinase/inhibitor combinations with no effect using small grey circles. For this, a dash "-" can be used for cases where no effect was observed or a threshold can be given, e.g., 3 μM as in the case of *Karaman et al. (2008)*. Kinase/inhibitor combination with a value above this threshold will be considered to have no effect. Lastly, the string "NA" needs to be used for data that is not available, in other words, a kinase/inhibitor combination that was not tested.

KR will produce one image for each inhibitor separately (each column of the input) or will produce plots with combination of inhibitors. In the later case, the user may select how many inhibitors are added to each combination plot up to a maximum of 6 inhibitors per combination plot. The combination plots are produced with consecutive columns in the input therefore the order of columns in the input must reflect the desired choice of ordering. Note that when displaying the data of more than one inhibitor per image, the option to annotate inhibitors with no effect is disabled. When running the procedure locally, a PostScript output will be created, which can be converted by the user to any other format as mentioned above. Using the inhibitor binding profiles procedure via the web interface allows the figures to be automatically converted to another format (PNG, JPG, PDF, TIFF). In any case, a KR input file is created for every figure, which allows them to be uploaded to the interface and further edited.

An example of an input file for this procedure (Example File 5 – Binding Data) is available online. This file contains the binding affinity for 3 ligands (Dasatinib, Erlotinib

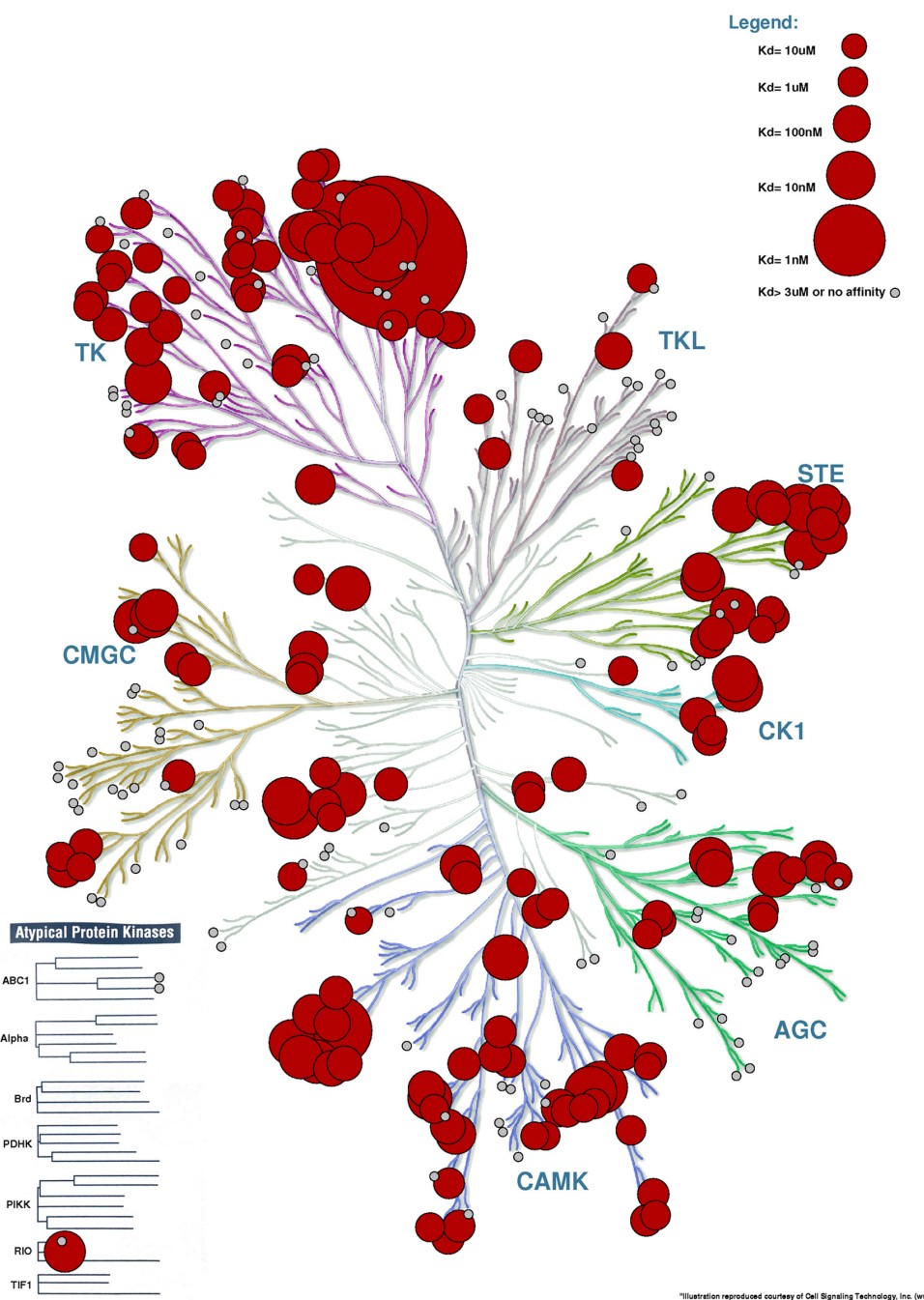

**Figure 5 Kinome Render used to display binding affinity for Sunitinib.** Recreation of the figure representing the affinity of Sunitinib for different kinases (*Karaman et al., 2008*). The kinases for which Sunitinib has an affinity <3 μM are annotated with a red circle. Other kinases tested but with an affinity >3 μM are shown in small grey circles. The bigger the circle, the higher the affinity.

and Sunitinib) (*Karaman et al., 2008*). Figure 6 displays the binding affinities projected on the human kinome tree from which we can see that Dasatinib mostly binds to Tyrosine Kinase family members while the binding profiles of Sunitinib and to a lesser extent Erlotinib are widespread across most human protein kinase families.

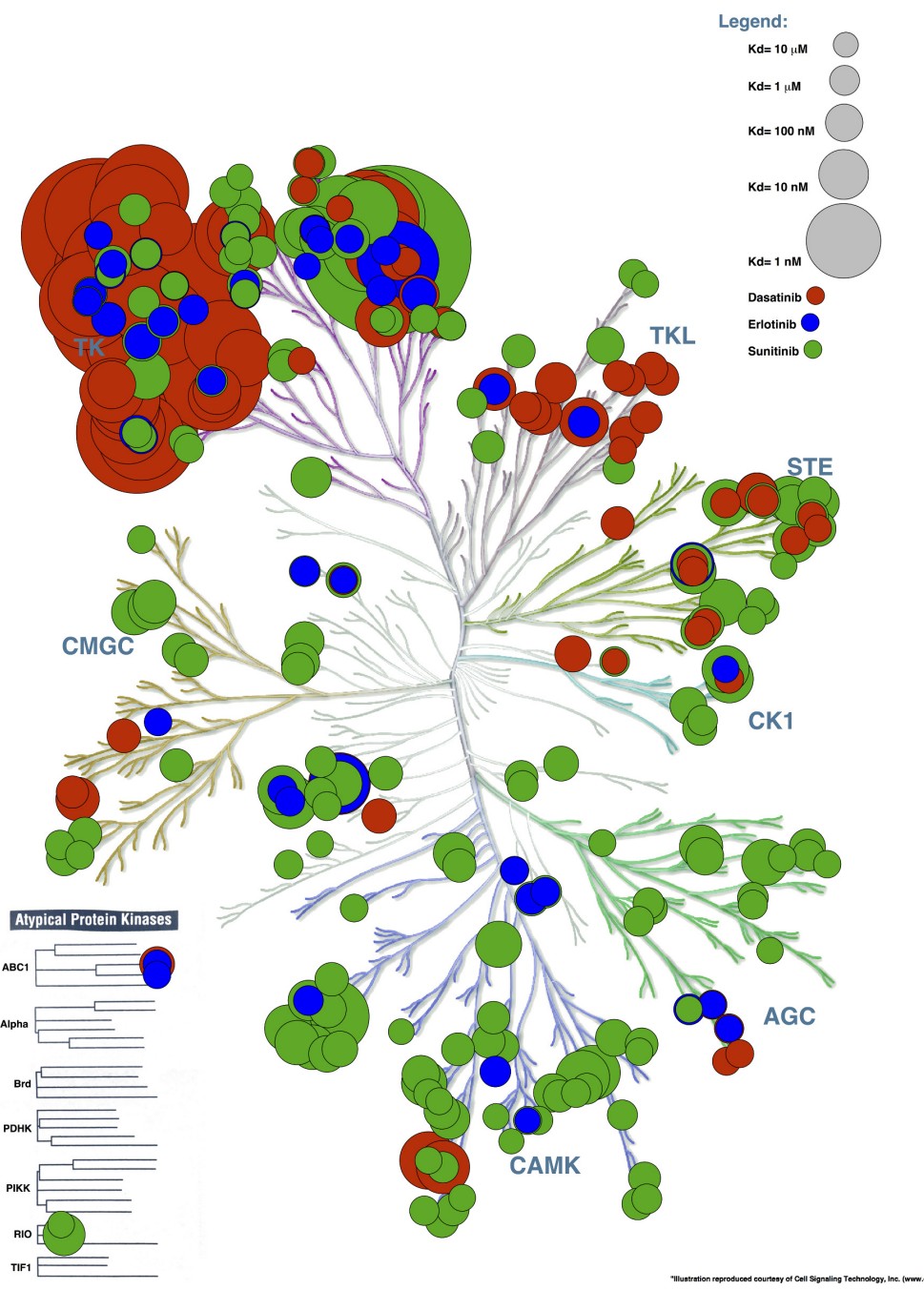

"Illustration reproduced courtesy of Cell Signaling Technology, Inc. (www.cellsignal.com)"

**Figure 6 Affinity of 3 small-molecules created with the inhibitor binding profile procedure.** Dasatinib binds primarily to TK family members whereas the inhibitor binding profiles of Erlotinib and to a lesser extent Sunitinib show widespread binding to human protein kinases throughout the entire family.

# CONCLUSIONS

Efficient visualization of data is a key step in every scientific study. Human protein kinases are relevant in a large number of biological processes and thus widely studied drug targets. The phylogenetic tree produced for the human protein kinase family has become an iconic

figure widely used in a large number of studies and can help reveal important trends. We have implemented a tool called Kinome Render that allows simple large-scale customized annotations of the human kinome tree. The method reduces the risk of errors prone to occur with hand made annotations. A user-friendly web interface was developed to make the tool easily accessible. We believe that Kinome Render can be a useful tool for anyone studying human protein kinases to quickly create annotated figures.

## Availability

Please visit http://bcb.med.usherbrooke.ca/kinomerender to access the interface or download the stand-alone version of Kinome Render. When publishing a figure obtained with KR, you must additionally acknowledge Cell Signaling Technology, Inc. (the creators of the original kinome tree image that is used by Kinome Render) in the following way: "Illustration reproduced courtesy of Cell Signaling Technology, Inc. (www.cellsignal.com)".

## ACKNOWLEDGEMENTS

The authors would like to thank María Inés Zylber for the critical reading and editing of the manuscript. The phylogenetic tree of the human kinome is reproduced with permission of Science and Cell Signaling Technology, Inc. R.J.N. is part of Centre de Recherche Clinique Étienne-Le Bel, a member of the Institute of Pharmacology of Sherbrooke, PROTEO (the Québec network for research on protein function, structure and engineering) and GRASP (Groupe de Recherche Axé sur la Structure des Protéines).

### Funding

No specific funding has been used in this study.

### Competing Interests

RJN is an Academic Editor for PeerJ. No other conflicts of interest exist.

### Author Contributions

- Matthieu Chartier performed the experiments, analyzed the data, contributed reagents/materials/analysis tools, wrote the paper.
- Thierry Chénard performed the experiments, analyzed the data, contributed reagents/materials/analysis tools, wrote the paper.
- Jonathan Barker performed the experiments, analyzed the data, contributed reagents/materials/analysis tools.
- Rafael Najmanovich conceived and designed the experiments, analyzed the data, wrote the paper.

## Data Deposition

The following information was supplied regarding the deposition of related data:

http://bcb.med.usherbrooke.ca/kinomerenderHelp.php

http://bcb.med.usherbrooke.ca/pages/Kinome/KinomeRender1.0.zip

http://bcb.med.usherbrooke.ca/files/TableS1.xls

## Supplemental Information

Supplemental information for this article can be found online at http://dx.doi.org/10.7717/peerj.126.

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
