# Peer review of "Kinome Render: a stand-alone and web-accessible tool to annotate the human protein kinome tree"

_PeerJ, doi:10.7717/peerj.126_

## Round 0.1 · original submission · Minor Revisions

You should foremost ensure that the tool works on all platforms, and that the usability and interface suggestions of the reviewers are carefully considered. Of these, batch upload of results at the very least is a must to make this tool useful for high-throughput omics studies. Reviewer 1 furthermore correctly points out that you will need to revise the formatting of your manuscript to comply with the PeerJ guidelines.

Reviewer 1 ·

Basic reporting

The authors described a new tool to visualize biological activities of kinase inhibitors onto a phylogenetic tree image. The manuscript is well written and scientific discussions are correctly argued. However the reference format does not follow the PeerJ policies, such as in-text citations and the reference section. Same comments for the figures presented in the manuscript and authors should carefully look at picture size and format here: https://peerj.com/about/author-instructions/#figures
This should be fixed before publication.

Experimental design

The work described in this manuscript will be valuable for the researchers working in kinase drug discovery, especially in data analysis from screening assays of kinase inhibitors. The investigation has been conducted rigorously and the methods is well explained. The standalone software provided by the authors works perfectly well.

Validity of the findings

No comments

Additional comments

I would advise the authors to differentiate between not inhibited and not tested kinases. They could use a grey sphere when kinases have not been screened (from the panel of 518 kinases) since it can be misleading with tiny circles, almost invisible, on the phylogenetic tree due to inactivity of a compound tested on those kinases

Reviewer 2 ·

Basic reporting

The article explains how to use Kinome Render, a newly developed application to facilitate the creation of annotated kinome trees. It is clearly written and details step by step the utilization of the tool. This type of tool is very useful for the easy visualization of kinase inhibitors profiles for instance, as explained by the authors.

The article however fails to provide references to the previous efforts in the design of such tools. The authors state that "some figures in these publications were created by hand in a laborious effort", which is true, but implies that some were not. And indeed, some tools are available. I believe it is of the duty of the authors to reference them. Commercial softwares such as TREEspot by Discoverex have to be named. The Human Kinome Java Component (http://tripod.nih.gov/?p=260) has to be taken into consideration and compared to. By minor modifications of the code (available on simple demand) this application can be used for similar purposes as Kinome Render (for instance http://pubs.acs.org/doi/abs/10.1021/pr301073j).

Experimental design

Whereas the existing tools only provide (to my knowledge) the possibility to annotate with one type of shapes, the diversity of annotations with Kinome Render has to be praised.

I have concerns concerning the practicality of the tool for large sets of data, which is the stated goal of the authors. The ticking of all the kinases on the web-interface or the copy pasting of the KR language lines for all the kinases is indeed tedious and should be made easier. A tab or comma delimited file as an input (solution chosen by the Human kinome java component for instance) would be immensely more practical as it could be easily generated from an excel file (or similar), which is the common output of many proteomics softwares. The advantages of filtering and sorting of excel could be used to annotate groups of kinases. As would be the vlookup function to convert IPI names of the kinases to the KR syntax accepted names.

If it is possible to modify the KR file to annotate the same kinase with more than one annotation (impossible with the web interface), these annotations are not displayed correctly. I see this as a drawback of the tool. I believe that 2 shapes should be superimposable so as to be able to compare 2 kinase inhibitors for instance. This would require some transparency for the shapes, an automatic ordering leading to "small on top", and an unlimited number of annotations (and not "a maximum of 523 annotations"), all properties implemented in the human kinome java component.

Another concern is the information that "annotations with the biggest scales should be written first so as not to hide smaller annotations in their vicinity". It is quite unrealistic to ask the user to sort his data in this way without him being able to use excel and its sorting possibilities for the generation of the input file.

It is surprising that the legend in the web-interface has to be written in the KR language whereas the authors state that "the web interface allows the creation of an annotated tree without any knowledge of the KR syntax". This should be amended to fulfill the goal of the authors.

Validity of the findings

It is of concern that the supplementary files provided cannot be directly loaded on the web-based tool, after opening them with wordpad. What should be the way to modify them?

Reviewer 3 ·

Basic reporting

No Comments

Experimental design

No Comments

Validity of the findings

The tool does not work with Kinome Render files created in Windows machines. The problem is that only Mac/Linux style line feeds are supported. Using files created on Windows results in numerous syntax errors. This has to be corrected before the manuscript can considered for publication.

The example files provided with the tool also do not work as these are not pure text files, and has the wrong line feeds.

With the correct line feeds the tool seems to work as wanted though.

Additional comments

The manuscript is well-written and contains the information required to use the tool. However the file format issues above have to be fixed to make the tool usable for Windows users.

The tool itself does not seem too complex, but should be very useful for researchers working with kinases, and makes it very easy to create production grade graphics for inclusion in scientific publications.

---

## Round 0.2 · accepted · Accept

I have no further comments. You addressed all reviewer concerns with your revision.

---

## Author Rebuttal · Round 0.2

## UNIVERSITÉ DE
## SHERBROOKE

**Département de Biochimie**
**Faculté de Médecine et des Sciences de la Santé**
3001, 12e Avenue Nord
Sherbrooke (Québec) - J1H 5N4
CANADA

**Rafael Najmanovich, Ph.D.**
Professeur Adjoint/Assistant Professor
Tel: 819-820-6868 ext. 12374
Fax: 819-564-5340
rafael.najmanovich@usherbrooke.ca

15 of July, 2013

Dear Dr. Martens,

We are happy to know that our work on the Kinome Render, a stand-alone and web-accessible tool to annotate the human protein kinome tree has been accepted with minor revisions. All reviewers gave extremely useful comments which we addressed in their entirety in the new version of our work. Please find in what follows a detailed description of each comment of the reviewers (bold) along with how we addressed each issue.

We hope our modifications will deem our submission accepted for publication.

Regards,

Rafael Najmanovich
(on behalf of all authors)

# Reviewer 1

**Basic reporting**
**The authors described a new tool to visualize biological activities of kinase inhibitors onto a phylogenetic tree image. The manuscript is well written and scientific discussions are correctly argued. However the reference format does not follow the PeerJ policies, such as in-text citations and the reference section. Same comments for the figures presented in the manuscript and authors should carefully look at picture size and format here: https://peerj.com/about/author-instructions/#figures This should be fixed before publication.**

The citations and figures have been formatted according to PeerJ policies. The DPI and size of each figure have been adjusted.
Fig 1 -> 1200 DPI
Fig 2 -> 600 DPI
Fig 3 -> 600 DPI
Fig 4 -> 1200 DPI
Fig 5 -> 1200 DPI
Fig 6 -> 1200 DPI

**Experimental design**
The work described in this manuscript will be valuable for the researchers working in kinase drug discovery, especially in data analysis from screening assays of kinase inhibitors. The investigation has been conducted rigorously and the methods is well explained. The standalone software provided by the authors works perfectly well.

**Validity of the findings**
**No comments**

**Comments for the author**
I would advise the authors to differentiate between not inhibited and not tested kinases. They could use a grey sphere when kinases have not been screened (from the panel of 518 kinases) since it can be misleading with tiny circles, almost invisible, on the phylogenetic tree due to inactivity of a compound tested on those kinases.

We now make a distinction between not inhibited and not tested. Kinases tested but with an affinity > 3µm are tagged with a grey spheres. The kinases not tested have no annotation. Figure 5 and its description have been modified.

# Reviewer 2

**Basic reporting**
The article explains how to use Kinome Render, a newly developed application to facilitate the creation of annotated kinome trees. It is clearly written and details step by step the utilization of the tool. This type of tool is very useful for the easy visualization of kinase inhibitors profiles for instance, as explained by the authors.

The article however fails to provide references to the previous efforts in the design of such tools. The authors state that "some figures in these publications were created by hand in a laborious effort", which is true, but implies that some were not. And indeed, some tools are available. I believe it is of the duty of the authors to reference them. Commercial softwares such as TREEspot by Discoverex have to be named. The Human Kinome Java Component (http://tripod.nih.gov/?p=260) has to be taken into consideration and compared to. By minor modifications of the code (available on simple demand) this application can be used for similar purposes as Kinome Render (for instance http://pubs.acs.org/doi/abs/10.1021/pr301073j).

The current manuscript now mentions the two other tools mentioned by the reviewer. Pros and cons in respect to Kinome Render are discussed.

**Experimental design**
Whereas the existing tools only provide (to my knowledge) the possibility to annotate with one type of shapes, the diversity of annotations with Kinome Render has to be praised.

**I have concerns concerning the practicality of the tool for large sets of data, which is the stated goal of the authors. The ticking of all the kinases on the web-interface or the copy pasting of the KR language lines for all the kinases is indeed tedious and should be made easier. A tab or comma delimited file as an input (solution chosen by the Human kinome java component for instance) would be immensely more practical as it could be easily generated from an excel file (or similar), which is the common output of many proteomics softwares.**

We added a new kinase selection method called «Batch upload». Users can now select kinases using kinase names, Uniprot or IPI in either comma, semicolon or new line separated strings. This allows to select kinases based on data retrieved from an Excel sheet. The manuscript and online «Get Started Guide» were modified accordingly.

Additionally, we created an entirely new way to input data into Kinome Render for the specific case of inhibition profiles using a tab delimited text file in a format that could be easily exported from excel or any other spread sheet software. The new inhibitor profile input method is described in a new section added to the manuscript.

**The advantages of filtering and sorting of excel could be used to annotate groups of kinases. As would be the vlookup function to convert IPI names of the kinases to the KR syntax accepted names.**

We added the IPI of each kinase in the database and they are now displayed in the tooltip.

**If it is possible to modify the KR file to annotate the same kinase with more than one annotation (impossible with the web interface), these annotations are not displayed correctly. I see this as a drawback of the tool.**

The web interface now allows to create as many annotations as the user wants. The same kinase can be used for any number of annotations.

**I believe that 2 shapes should be superimposable so as to be able to compare 2 kinase inhibitors for instance. This would require some transparency for the shapes, an automatic ordering leading to "small on top", and an unlimited number of annotations (and not "a maximum of 523 annotations"), all properties implemented in the human kinome java component.**

Automatic ordering was already implemented in the web interface. We implemented it in the stand-alone version. We also removed all limitations on the number of annotations, therefore it is now possible to add as many annotations as desired and made the corresponding changes to the manuscript to clarify and address these points.

**Another concern is the information that "annotations with the biggest scales should be written first so as not to hide smaller annotations in their vicinity". It is quite**

**unrealistic to ask the user to sort his data in this way without him being able to use excel and its sorting possibilities for the generation of the input file.**

As mentioned above, automatic ordering was already implemented in the web interface. We implemented it in the stand-alone version.

**It is surprising that the legend in the web-interface has to be written in the KR language whereas the authors state that "the web interface allows the creation of an annotated tree without any knowledge of the KR syntax". This should be amended to fulfill the goal of the authors.**

The number of possible different types of annotations implemented in Kinome Render leads to an exponential number of possible combinations. It would be almost impossible to make the FULLY automatic creation of aesthetically pleasing legends without restricting the possibilities offered to the user in kinome render. We therefore are of the opinion that it is acceptable ask of the users to learn some very basic KR syntax commands to create legends that fully describe any possible choice of representation permitted in Kinome Render. Furthermore, the creation of legends is optional, therefore, a user that prefers to create the legend independently may do so. However, to further simplify the syntax, predefined colors are now supported in both versions of Kinome Render in addition to RGB values.

**Validity of the findings**
**It is of concern that the supplementary files provided cannot be directly loaded on the web-based tool, after opening them with wordpad. What should be the way to modify them?**

Kinome Render and the web interface now support Windows, Mac and Linux line feeds. Any file in plain text can now be used as input for the 3 platforms. It is important to note the PeerJ web interface does not allow the uploading of plain text files. Therefore, we decided to provide access to the plain text example files via the Kinome Render interface and not as supplementary material online via PeerJ as originally done as such files could not be used directly as input files for Kinome Render. The relevant sections of the manuscript were changed to reflect this point.

# Reviewer 3

**Basic reporting**
**No Comments**

**Experimental design**
**No Comments**

**Validity of the findings**

**The tool does not work with Kinome Render files created in Windows machines. The problem is that only Mac/Linux style line feeds are supported. Using files created on Windows results in numerous syntax errors. This has to be corrected before the manuscript can considered for publication.**

As mentioned above, Kinome Render and the web interface now support Windows, Mac and Linux line feeds. Any file in plain text can now be used as input for the 3 platforms.

**The example files provided with the tool also do not work as these are not pure text files, and has the wrong line feeds. With the correct line feeds the tool seems to work as wanted though.**

As described above, PeerJ does not allow the upload of plain text files. Therefore, the example files in a ready-to-use plain text format are provided directly online as mentioned above.

**Comments for the author**
**The manuscript is well-written and contains the information required to use the tool. However the file format issues above have to be fixed to make the tool usable for Windows users. The tool itself does not seem too complex, but should be very useful for researchers working with kinases, and makes it very easy to create production grade graphics for inclusion in scientific publications.**

The issue of file format has been fully resolved (see above).

Rafael Najmanovich